# PRNet: Self-Supervised Learning for Partial-to-Partial Registration

**Yue Wang**
Massachusetts Institute of Technology
`yuewangx@mit.edu`

**Justin Solomon**
Massachusetts Institute of Technology
`jsolomon@mit.edu`

## Abstract

We present a simple, flexible, and general framework titled Partial Registration Network (PRNet), for partial-to-partial point cloud registration. Inspired by recently-proposed learning-based methods for registration, we use deep networks to tackle non-convexity of the alignment and partial correspondence problems. While previous learning-based methods assume the entire shape is visible, PRNet is suitable for partial-to-partial registration, outperforming PointNetLK, DCP, and non-learning methods on synthetic data. PRNet is self-supervised, jointly learning an appropriate geometric representation, a keypoint detector that finds points in common between partial views, and keypoint-to-keypoint correspondences. We show PRNet predicts keypoints and correspondences consistently across views and objects. Furthermore, the learned representation is transferable to classification.

## 1 Introduction

*Registration* is the problem of predicting a rigid motion aligning one point cloud to another. Algorithms for this task have steadily improved, using machinery from vision, graphics, and optimization. These methods, however, are usually orders of magnitude slower than "vanilla" Iterative Closest Point (ICP), and some have hyperparameters that must be tuned case-by-case. The trade-off between efficiency and effectiveness is steep, reducing generalizability and/or practicality.

Recently, PointNetLK [1] and Deep Closest Point (DCP) [2] show that learning-based registration can be faster and more robust than classical methods, even when trained on different datasets. These methods, however, cannot handle partial-to-partial registration, and their one-shot constructions preclude refinement of the predicted alignment.

We introduce the Partial Registration Network (PRNet), a sequential decision-making framework designed to solve a broad class of registration problems. Like ICP, our method is designed to be applied *iteratively*, enabling coarse-to-fine refinement of an initial registration estimate. A critical new component of our framework is a keypoint detection sub-module, which identifies points that match in the input point clouds based on co-contextual information. Partial-to-partial point cloud registration then boils down to detecting keypoints the two point clouds have in common, matching these keypoints to one another, and solving the Procrustes problem.

Since PRNet is designed to be applied iteratively, we use Gumbel–Softmax [3] with a straight-through gradient estimator to sample keypoint correspondences. This new architecture and learning procedure modulates the sharpness of the matching; distant point clouds given to PRNet can be coarsely matched using a diffuse (fuzzy) matching, while the final refinement iterations prefer sharper maps. Rather than introducing another hyperparameter, PRNet uses a sub-network to predict the temperature [4] of the Gumbel–Softmax correspondence, which can be cast as a simplified version of the actor-critic method. That is, PRNet *learns* to modulate the level of map sharpness each time it is applied.

We train and test PRNet on ModelNet40 and on real data. We visualize the keypoints and correspondences for shapes from the same or different categories. We transfer the learned representations to shape classification using a linear SVM, achieving comparable performance to state-of-the-art supervised methods on ModelNet40.

**Contributions.** We summarize our key contributions as follows:

- We present the *Partial Registration Network* (PRNet), which enables partial-to-partial point cloud registration using deep networks with state-of-the-art performance.
- We use Gumbel–Softmax with straight-through gradient estimation to obtain a sharp and near-differentiable mapping function.
- We design an *actor-critic closest point* module to modulate the sharpness of the correspondence using an action network and a value network. This module predicts more accurate rigid transformations than differentiable soft correspondence methods with fixed parameters.
- We show registration is a useful proxy task to learn representations for 3D shapes. Our representations can be transferred to other tasks, including keypoint detection, correspondence prediction, and shape classification.
- We release our code to facilitate reproducibility and future research.

## 2  Related Work

**Rigid Registration.** ICP [5] and variants [6, 7, 8, 9] have been widely used for registration. Recently, probabilistic models [10, 11, 12] have been proposed to handle uncertainty and partiality. Another trend is to improve the optimization: [13] applies Levenberg—Marquardt to the ICP objective, while global methods seek a solution using branch-and-bound [14], Riemannian optimization [15], convex relaxation [16], mixed-integer programming [17], and semidefinite programming [18].

**Learning on Point Clouds and 3D Shapes.** Deep Sets [19] and PointNet [20] pioneered deep learning on point sets, a challenge problem in learning and vision. These methods take coordinates as input, embed them to high-dimensional space using shared multilayer perceptrons (MLPs), and use a symmetric function (e.g., $\max$ or $\sum$) to aggregate features. Follow-up works incorporate local information, including PointNet++ [21], DGCNN [22], PointCNN [23], and PCNN [24]. Another branch of 3D learning designs convolution-like operations for shapes or applies graph convolutional networks (GCNs) [25, 26] to triangle meshes [27, 28], exemplifying architectures on non-Euclidean data termed *geometric deep learning* [29]. Other works, including SPLATNet [30], SplineCNN [31], KPConv [32], and GWCNN [33], transform 3D shapes to regular grids for feature learning.

**Keypoints and Correspondence.** Correspondence and registration are dual tasks. Correspondence is the approach while registration is the output, or vice versa. Countless efforts tackle the correspondence problem, either at the point-to-point or part-to-part level. Due to the $O(n^2)$ complexity of point-to-point correspondence matrices and $O(n!)$ possible permutations, most methods (e.g., [34, 35, 36, 37, 38, 39, 40, 41]) compute a sparse set of correspondences and extend them to dense maps, often with bijectivity as an assumption or regularizer. Other efforts use more exotic representations of correspondences. For example, functional maps [42] generalize to mappings between functions on shapes rather than points on shapes, expressing a map as a linear operator in the Laplace–Beltrami eigenbasis. Mathematical methods like functional maps can be made 'deep' using priors learned from data: Deep functional maps [43, 44] learn descriptors rather than designing them by hand.

For partial-to-partial registration, we cannot compute bijective correspondences, invalidating many past representations. Instead, keypoint detection is more secure. To extract a sparser representation, KeyPointNet [45] uses registration and multiview consistency as supervision to learn a keypoint detector on 2D images; our method performs keypoint detection on point clouds. In contrast to our model, which learns correspondences from registration, [46] uses correspondence prediction as the training objective to learn how to segment parts. In particular, it utilizes PointNet++ [21] to product point-wise features, generates matching using a correspondence proposal module, and finally trains the pipeline with ground-truth correspondences.

**Self-supervised Learning.** Humans learn knowledge not only from teachers but also by predicting and reasoning about unlabeled information. Inspired by this observation, self-supervised learning usually involves predicting part of an input from another part [47, 48], solving one task using features

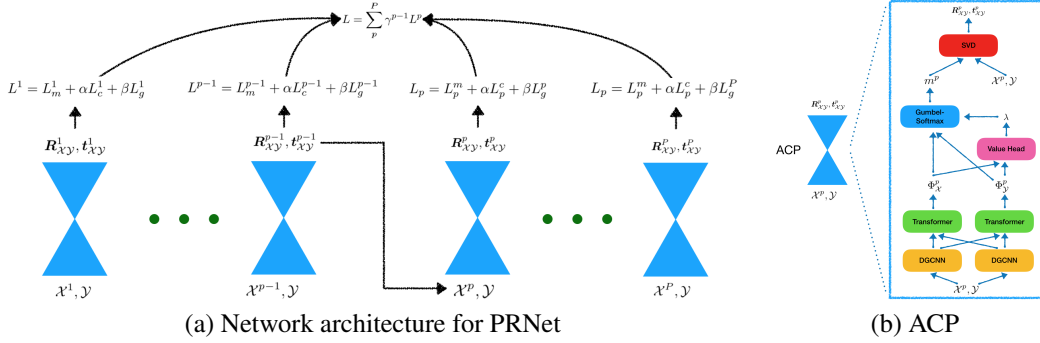

(a) Network architecture for PRNet       (b) ACP

Figure 1: Network architecture for PRNet and ACP.

learned from another task [45] and/or enforcing consistency from different views/modalities [49, 50]. Self-supervised pretraining is an effective way to transfer knowledge learned from massive unlabeled data to tasks where labeled data is limited. For example, BERT [51] surpasses state-of-the-art in natural language processing by learning from contextual information. ImageNet Pretrain [52] commonly provides initialization for vision tasks. Video-audio joint analysis [53, 54, 55] utilizes modality consistency to learn representations. Our method is also self-supervised, in the sense that no labeled data is needed.

**Actor–Critic Methods.** Many recent works can be counted as actor–critic methods, including deep reinforcement learning [56], generative modeling [57], and sequence generation [58]. These methods generally involve two functions: taking actions and estimating values. The predicted values can be used to improve the actions while the values are collected when the models interact with environment. PRNet uses a sub-module (value head) to predict the level of granularity at which we should map two shapes. The value adjusts the temperature of Gumbel–Softmax in the action head.

## 3 Method

We establish preliminaries about the rigid alignment problem and related algorithms in §3.1; then, we present PRNet in §3.2. For ease of comparison to previous work, we use the same notation as [2].

### 3.1 Preliminaries: Registration, ICP, and DCP

Consider two point clouds $\mathcal{X} = \{\boldsymbol{x}_1, \ldots, \boldsymbol{x}_i, \ldots, \boldsymbol{x}_N\} \subset \mathbb{R}^3$ and $\mathcal{Y} = \{\boldsymbol{y}_1, \ldots, \boldsymbol{y}_j, \ldots, \boldsymbol{y}_M\} \subset \mathbb{R}^3$. The basic task in rigid registration is to find a rotation $\boldsymbol{R}_{\mathcal{X}\mathcal{Y}}$ and translation $\boldsymbol{t}_{\mathcal{X}\mathcal{Y}}$ that rigidly align $\mathcal{X}$ to $\mathcal{Y}$. When $M = N$, ICP and its peers approach this task by minimizing the objective function

$$E(\boldsymbol{R}_{\mathcal{X}\mathcal{Y}}, \boldsymbol{t}_{\mathcal{X}\mathcal{Y}}, m) = \frac{1}{N} \sum_{i=1}^{N} \|\boldsymbol{R}_{\mathcal{X}\mathcal{Y}}\boldsymbol{x}_i + \boldsymbol{t}_{\mathcal{X}\mathcal{Y}} - \boldsymbol{y}_{m(x_i)}\|^2. \tag{1}$$

Here, the rigid transformation is defined by a pair $[\boldsymbol{R}_{\mathcal{X}\mathcal{Y}}, \boldsymbol{t}_{\mathcal{X}\mathcal{Y}}]$, where $\boldsymbol{R}_{\mathcal{X}\mathcal{Y}} \in \mathrm{SO}(3)$ and $\boldsymbol{t}_{\mathcal{X}\mathcal{Y}} \in \mathbb{R}^3$; $m$ maps from points in $\mathcal{X}$ to points in $\mathcal{Y}$. Assuming $m$ is fixed, the alignment in (1) is given in closed-form by

$$\boldsymbol{R}_{\mathcal{X}\mathcal{Y}} = \boldsymbol{V}\boldsymbol{U}^\top \qquad \text{and} \qquad \boldsymbol{t}_{\mathcal{X}\mathcal{Y}} = -\boldsymbol{R}_{\mathcal{X}\mathcal{Y}}\overline{\boldsymbol{x}} + \overline{\boldsymbol{y}}, \tag{2}$$

where $\boldsymbol{U}$ and $\boldsymbol{V}$ are obtained using the singular value decomposition (SVD) $\boldsymbol{H} = \boldsymbol{U}\boldsymbol{S}\boldsymbol{V}^\top$, with $\boldsymbol{H} = \sum_{i=1}^{N}(\boldsymbol{x}_i - \overline{\boldsymbol{x}})(\boldsymbol{y}_{m(x_i)} - \overline{\boldsymbol{y}})^\top$. In this expression, centroids of $\mathcal{X}$ and $\mathcal{Y}$ are defined as $\overline{\boldsymbol{x}} = \frac{1}{N}\sum_{i=1}^{N}\boldsymbol{x}_i$ and $\overline{\boldsymbol{y}} = \frac{1}{N}\sum_{i=1}^{N}\boldsymbol{y}_{m(x_i)}$, respectively.

We can understand ICP and the more recent learning-based DCP method [2] as providing different choices of $m$:

**Iterative Closest Point.** ICP chooses $m$ to minimize (1) with $[\boldsymbol{R}_{\mathcal{X}\mathcal{Y}}, \boldsymbol{t}_{\mathcal{X}\mathcal{Y}}]$ fixed, yielding:

$$m(\boldsymbol{x}_i, \mathcal{Y}) = \arg\min_{j} \|\boldsymbol{R}_{\mathcal{X}\mathcal{Y}}\boldsymbol{x}_i + \boldsymbol{t}_{\mathcal{X}\mathcal{Y}} - \boldsymbol{y}_j\|_2 \tag{3}$$

ICP approaches a fixed point by alternating between (2) and (3); each step decreases the objective (1). Since (1) is non-convex, however, there is no guarantee that ICP reaches a global optimum.

**Deep Closest Point.** DCP uses deep networks to learn $m$. In this method, $\mathcal{X}$ and $\mathcal{Y}$ are embedded using learned functions $\mathcal{F}_\mathcal{X}$ and $\mathcal{F}_\mathcal{Y}$ defined by a Siamese DGCNN [22]; these lifted point clouds are optionally contextualized by a Transformer module [59], yielding embeddings $\Phi_\mathcal{X}$ and $\Phi_\mathcal{Y}$. The mapping $m$ is then

$$m(\boldsymbol{x}_i, \mathcal{Y}) = \text{softmax}(\Phi_\mathcal{Y} \Phi_{\boldsymbol{x}_i}^\top). \tag{4}$$

This formula is applied in one shot followed by (2) to obtain the rigid alignment. The loss used to train this pipeline is mean-squared error (MSE) between ground-truth rigid motion from synthetically-rotated point clouds and prediction; the network is trained end-to-end.

## 3.2 Partial Registration Network

DCP is a one-shot algorithm, in that a single pass through the network determines the output for each prediction task. Analogously to ICP, PRNet is designed to be *iterative*; multiple passes of a point cloud through PRNet refine the alignment. The steps of PRNet, illustrated in Figure 1, are as follows:

1. take as input point clouds $\mathcal{X}$ and $\mathcal{Y}$;

2. detect keypoints of $\mathcal{X}$ and $\mathcal{Y}$;

3. predict a mapping from keypoints of $\mathcal{X}$ to keypoints of $\mathcal{Y}$;

4. predict a rigid transformation $[\boldsymbol{R}_\mathcal{XY}, \boldsymbol{t}_\mathcal{XY}]$ aligning $\mathcal{X}$ to $\mathcal{Y}$ based on the keypoints and map;

5. transform $\mathcal{X}$ using the obtained transformation;

6. return to 1 using the pair $(\boldsymbol{R}_\mathcal{XY}\mathcal{X} + \boldsymbol{t}_\mathcal{XY}, \mathcal{Y})$ as input.

When predicting a mapping from keypoints in $\mathcal{X}$ to keypoints in $\mathcal{Y}$, PRNet uses Gumbel–Softmax [3] to sample a matching matrix, which is sharper than (4) and approximately differentiable. It has a value network to predict a temperature for Gumbel–Softmax, so that the whole framework can be seen as an actor-critic method. We present details of and justifications behind the design below.

**Notation.** Denote by $\mathcal{X}^p = \{\boldsymbol{x}_1^p, \ldots, \boldsymbol{x}_i^p, \ldots, \boldsymbol{x}_N^p\}$ the rigid motion of $\mathcal{X}$ to align to $\mathcal{Y}$ after $p$ applications of PRNet; $\mathcal{X}^1$ and $\mathcal{Y}^1$ are initial input shapes. We will use $[\boldsymbol{R}_\mathcal{XY}^p, \boldsymbol{t}_\mathcal{XY}^p]$ to denote the $p$-th rigid motion predicted by PRNet for the input pair $(\mathcal{X}, \mathcal{Y})$.

Since our training pairs are synthetically generated, before applying PRNet we know the ground-truth $[\boldsymbol{R}_\mathcal{XY}^*, \boldsymbol{t}_\mathcal{XY}^*]$ aligning $\mathcal{X}$ to $\mathcal{Y}$. From these values, during training we can compute "local" ground-truth $[\boldsymbol{R}_\mathcal{XY}^{p*}, \boldsymbol{t}_\mathcal{XY}^{p*}]$ on-the-fly, which maps the current $(\mathcal{X}^p, \mathcal{Y})$ to the best alignment:

$$\boldsymbol{R}_\mathcal{XY}^{p*} = \boldsymbol{R}_\mathcal{XY}^* \boldsymbol{R}_\mathcal{XY}^{1\ldots p\top} \quad \text{and} \quad \boldsymbol{t}_\mathcal{XY}^{p*} = \boldsymbol{t}_\mathcal{XY}^* - \boldsymbol{R}_\mathcal{XY}^{p*} \boldsymbol{t}_\mathcal{XY}^{1\ldots p}, \tag{5}$$

where

$$\boldsymbol{R}_\mathcal{XY}^{1\ldots p} = \boldsymbol{R}_\mathcal{XY}^{p-1} \ldots \boldsymbol{R}_\mathcal{XY}^1 \quad \text{and} \quad \boldsymbol{t}_\mathcal{XY}^{1\ldots p} = \boldsymbol{R}_\mathcal{XY}^{p-1} \boldsymbol{t}_\mathcal{XY}^{1\ldots p-1} + \boldsymbol{t}_\mathcal{XY}^{p-1}. \tag{6}$$

We use $m^p$ to denote the mapping function in $p$-th step.

Synthesizing the notation above, $\mathcal{X}^p$ is given by

$$\boldsymbol{x}_i^p = \boldsymbol{R}_\mathcal{XY}^{p-1} \boldsymbol{x}_i^{p-1} + \boldsymbol{t}_\mathcal{XY}^{p-1} \tag{7}$$

where

$$\boldsymbol{R}_\mathcal{XY}^p = \boldsymbol{V}^p \boldsymbol{U}^{p\top} \quad \text{and} \quad \boldsymbol{t}_\mathcal{XY}^p = -\boldsymbol{R}_\mathcal{XY}^p \overline{\boldsymbol{x}}^p + \overline{\boldsymbol{y}}. \tag{8}$$

In this equation, $\boldsymbol{U}^p$ and $\boldsymbol{V}^p$ are computed using (2) from $\mathcal{X}^p$, $\mathcal{Y}^p$, and $m^p$.

**Keypoint Detection.** For partial-to-partial registration, usually $N \neq M$ and only subsets of $\mathcal{X}$ and $\mathcal{Y}$ match to one another. To detect these mutually-shared patches, we design a simple yet efficient keypoint detection module based on the observation that the $L^2$ norms of features tend to indicate whether a point is important.

Using $\mathcal{X}_k^p$ and $\mathcal{Y}_k^p$ to denote the $k$ keypoints for $\mathcal{X}^p$ and $\mathcal{Y}^p$, we take

$$\begin{aligned} \mathcal{X}_k^p &= \mathcal{X}^p(\text{topk}(\|\Phi_{\boldsymbol{x}_1}^p\|_2, \ldots, \|\Phi_{\boldsymbol{x}_i}^p\|_2, \ldots, \|\Phi_{\boldsymbol{x}_N}^p\|_2)) \\ \mathcal{Y}_k^p &= \mathcal{Y}^p(\text{topk}(\|\Phi_{\boldsymbol{y}_1}^p\|_2, \ldots, \|\Phi_{\boldsymbol{y}_i}^p\|_2, \ldots, \|\Phi_{\boldsymbol{y}_M}^p\|_2)) \end{aligned} \tag{9}$$

where $\mathrm{topk}(\cdot)$ extracts the indices of the $k$ largest elements of the given input. Here, $\Phi$ denotes embeddings learned by DGCNN and Transformer.

By aligning only the keypoints, we remove irrelevant points from the two input clouds that are not shared in the partial correspondence. In particular, we can now solve the Procrustes problem that matches keypoints of $\mathcal{X}$ and $\mathcal{Y}$. We show in §4.3 that although we do not provide explicit supervision, PRNet still learns how to detect keypoints reasonably.

**Gumbel–Softmax Sampler.** One key observation in ICP and DCP is that (3) usually is not differentiable with respect to the map $m$ but by definition yields a sharp correspondence between the points in $\mathcal{X}$ and the points in $\mathcal{Y}$. In contrast, the smooth function (4) in DCP is differentiable, but in exchange for this differentiability the mapping is blurred. We desire the best of both worlds: A potentially sharp mapping function that admits backpropagation.

To that end, we use Gumbel–Softmax [3] to sample a matching matrix. Using a straight-through gradient estimator, this module is approximately differentiable. In particular, the Gumbel–Softmax mapping function is given by

$$m^p(\boldsymbol{x}_i, \mathcal{Y}) = \mathrm{one\,hot}\left[\arg\max_j \mathrm{softmax}(\Phi^p_{\mathcal{Y}} \Phi^{p\top}_{\boldsymbol{x}_i} + g_{ij})\right], \tag{10}$$

where $(g_{i1}, \ldots, g_{ij}, \ldots, g_{iN})$ are i.i.d. samples drawn from Gumbel(0, 1). The map in (10) is not differentiable due to the discontinuity of $\arg\max$, but the straight-through gradient estimator [60] yields (biased) subgradient estimates with low variance. Following their methodology, on backward evaluation of the computational graph, we use (4) to compute $\frac{\partial L}{\partial \Phi^p_*}$, ignoring the $\mathrm{one\,hot}$ operator and the $\arg\max$ term.

**Actor-Critic Closest Point (ACP).** The mapping functions (4) and (10) have fixed "temperatures," that is, there is no control over the sharpness of the mapping matrix $m^p$. In PRNet, we wish to adapt the sharpness of the map based on the alignment of the two shapes. In particular, for low values of $p$ (the initial iterations of alignment) we may satisfied with high-entropy approximate matchings that obtain a coarse alignment; later during iterative evaluations, we can sharpen the map to align individual pairs of points.

To make this intuition compatible with PRNet's learning-based architecture, we add a parameter $\lambda$ to (10) to yield a generalized Gumbel–Softmax matching matrix:

$$m^p(\boldsymbol{x}_i, \mathcal{Y}) = \mathrm{one\,hot}\left[\arg\max_j \mathrm{softmax}\left(\frac{\Phi^p_{\mathcal{Y}} \Phi^{p\top}_{\boldsymbol{x}_i} + g_{ij}}{\lambda}\right)\right] \tag{11}$$

When $\lambda$ is large, the map matrix $m^p$ is smoothed out; as $\lambda \to 0$ the map approaches a binary matrix.

It is difficult to choose a single $\lambda$ that suffices for all $(\mathcal{X}, \mathcal{Y})$ pairs; rather, we wish $\lambda$ to be chosen adaptively and automatically to extract the best alignment for each pair of point clouds. Hence, we use a small network $\Theta$ to predict $\lambda$ based on global features $\Psi^p_{\mathcal{X}}$ and $\Psi^p_{\mathcal{Y}}$ aggregated from $\Phi^p_{\mathcal{X}}$ and $\Phi^p_{\mathcal{Y}}$ channel-wise by global pooling (averaging). In particular, we take $\lambda = \Theta(\Psi^p_{\mathcal{X}}, \Psi^p_{\mathcal{Y}})$, where $\Psi^p_{\mathcal{X}} = \mathrm{avg}_i \Phi^p_{\boldsymbol{x}_i}$ and $\Psi^p_{\mathcal{Y}} = \mathrm{avg}_i \Phi^p_{\boldsymbol{y}_i}$. In the parlance of reinforcement learning, this choice can be seen as a simplified version of actor-critic method. $\Phi^p_{\mathcal{X}}$ and $\Phi^p_{\mathcal{Y}}$ are learned jointly with DGCNN [22] and Transformer [59]; then an actor head outputs a rigid motion, where (11) uses the $\lambda$ predicted from a critic head.

**Loss Function.** The final loss $L$ is the summation of several terms $L_p$, indexed by the number $p$ of passes through PRNet for the input pair. $L_p$ consists of three terms: a rigid motion loss $L^m_p$, a cycle consistency loss $L^c_p$, and a global feature alignment loss $L_g$. We also introduce a discount factor $\gamma < 1$ to promote alignment within the first few passes through PRNet; during training we pass each input pair through PRNet $P$ times.

Combining the terms above, we have

$$L = \sum_{p=1}^{P} \gamma^{p-1} L_p, \qquad \text{where} \qquad L_p = L^m_p + \alpha L^c_p + \beta L^p_g. \tag{12}$$

The rigid motion loss $L^m_p$ is,

$$L^m_p = \|\boldsymbol{R}^{p\top}_{\mathcal{X}\mathcal{Y}} \boldsymbol{R}^{p*}_{\mathcal{X}\mathcal{Y}} - I\|^2 + \|\boldsymbol{t}^p_{\mathcal{X}\mathcal{Y}} - \boldsymbol{t}^{p*}_{\mathcal{X}\mathcal{Y}}\|^2 \tag{13}$$

| Model | MSE($\boldsymbol{R}$)↓ | RMSE($\boldsymbol{R}$)↓ | MAE($\boldsymbol{R}$)↓ | R²($\boldsymbol{R}$)↑ | MSE($\boldsymbol{t}$)↓ | RMSE($\boldsymbol{t}$)↓ | MAE($\boldsymbol{t}$)↓ | R²($\boldsymbol{t}$)↑ |
|---|---|---|---|---|---|---|---|---|
| ICP | 1134.552 | 33.683 | 25.045 | -5.696 | 0.0856 | 0.293 | 0.250 | -0.037 |
| Go-ICP [14] | 195.985 | 13.999 | 3.165 | -0.157 | 0.0011 | 0.033 | 0.012 | 0.987 |
| FGR [61] | 126.288 | 11.238 | 2.832 | 0.256 | 0.0009 | 0.030 | 0.008 | 0.989 |
| PointNetLK [1] | 280.044 | 16.735 | 7.550 | -0.654 | 0.0020 | 0.045 | 0.025 | 0.975 |
| DCP-v2 [2] | 45.005 | 6.709 | 4.448 | 0.732 | 0.0007 | 0.027 | 0.020 | 0.991 |
| PRNet (Ours) | **10.235** | **3.199257** | **1.454** | **0.939** | **0.0003** | **0.016** | **0.010** | **0.997** |

Table 1: Test on unseen point clouds

| Model | MSE($\boldsymbol{R}$)↓ | RMSE($\boldsymbol{R}$)↓ | MAE($\boldsymbol{R}$)↓ | R²($\boldsymbol{R}$)↑ | MSE($\boldsymbol{t}$)↓ | RMSE($\boldsymbol{t}$)↓ | MAE($\boldsymbol{t}$)↓ | R²($\boldsymbol{t}$)↑ |
|---|---|---|---|---|---|---|---|---|
| ICP | 1217.618 | 34.894 | 25.455 | -6.253 | 0.086 | 0.293 | 0.251 | -0.038 |
| Go-ICP [14] | 157.072 | 12.533 | 2.940 | 0.063 | 0.0009 | 0.031 | 0.010 | 0.989 |
| FGR [61] | 98.635 | 9.932 | **1.952** | 0.414 | 0.0014 | 0.038 | 0.007 | 0.983 |
| PointNetLK [1] | 526.401 | 22.943 | 9.655 | -2.137 | 0.0037 | 0.061 | 0.033 | 0.955 |
| DCP-v2 [2] | 95.431 | 9.769 | 6.954 | 0.427 | 0.0010 | 0.034 | 0.025 | 0.986 |
| PRNet (Ours) | **24.857** | **4.986** | 2.329 | **0.850** | **0.0004** | **0.021** | **0.015** | **0.995** |
| PRNet (Ours*) | **15.624** | **3.953** | **1.712** | **0.907** | **0.0003** | **0.017** | **0.011** | **0.996** |

Table 2: Test on unseen categories. PRNet (Ours*) denotes the model trained on ShapeNetCore and tested on ModelNet40 held-out categories. Others are trained on first 20 ModelNet40 categories and tested on ModelNet40 held-out categories.

Equation (5) gives the "localized" ground truth values for $\boldsymbol{R}^{p*}_{\mathcal{XY}}, \boldsymbol{t}^{p*}_{\mathcal{XY}}$. Denoting the rigid motion from $\mathcal{Y}$ to $\mathcal{X}$ in step $p$ as $[\boldsymbol{R}^p_{\mathcal{YX}}, \boldsymbol{t}^p_{\mathcal{YX}}]$, the cycle consistency loss is

$$L^c_p = \|\boldsymbol{R}^p_{\mathcal{XY}}\boldsymbol{R}^p_{\mathcal{YX}} - I\|^2 + \|\boldsymbol{t}^p_{\mathcal{XY}} - \boldsymbol{t}^p_{\mathcal{YX}}\|^2. \tag{14}$$

Our last loss term is a global feature alignment loss, which enforces alignment of global features $\Psi^p_{\mathcal{X}}$ and $\Psi^p_{\mathcal{Y}}$. Mathematically, the global feature alignment loss is

$$L^p_g = \|\Psi^p_{\mathcal{X}} - \Psi^p_{\mathcal{Y}}\|. \tag{15}$$

This global feature alignment loss also provides signal for determining $\lambda$. When two shapes are close in global feature space, $\lambda$ should be small, yielding a sharp matching matrix; when two shapes are far from each other, $\lambda$ increases and the map is blurry.

## 4 Experiments

Our experiments are divided into four parts. First, we show performance of PRNet on a partial-to-partial registration task on synthetic data in §4.1. Then, we show PRNet can generalize to real data in §4.2. Third, we visualize the keypoints and correspondences predicted by PRNet in §4.3. Finally, we show a linear SVM trained on representations learned by PRNet can achieve comparable results to supervised learning methods in §4.4.

### 4.1 Partial-to-Partial Registration on ModelNet40

We evaluate partial-to-partial registration on ModelNet40 [62]. There are 12,311 CAD models spanning 40 object categories, split to 9,843 for training and 2,468 for testing. Point clouds are sampled from the CAD models by farthest-point sampling on the surface. During training, a point cloud with 1024 points $\mathcal{X}$ is sampled. Along each axis, we randomly draw a rigid transformation; the rotation along each axis is sampled in $[0, 45°]$ and translation is in $[-0.5, 0.5]$. We apply the rigid transformation to $\mathcal{X}$, leading to $\mathcal{Y}$. We simulate partial scans of $\mathcal{X}$ and $\mathcal{Y}$ by randomly placing a point in space and computing its 768 nearest neighbors in $\mathcal{X}$ and $\mathcal{Y}$ respectively.

We measure mean squared error (MSE), root mean squared error (RMSE), mean absolute error (MAE), and coefficient of determination (R²). Angular measurements are in units of degrees. MSE, RMSE and MAE should be zero while R² should be one if the rigid alignment is perfect. We compare our model to ICP, Go-ICP [14], Fast Global Registration (FGR) [61], and DCP [2].

| Model | MSE($R$)↓ | RMSE($R$)↓ | MAE($R$)↓ | R$^2$($R$)↑ | MSE($t$)↓ | RMSE($t$)↓ | MAE($t$)↓ | R$^2$($t$)↑ |
|---|---|---|---|---|---|---|---|---|
| ICP | 1229.670 | 35.067 | 25.564 | -6.252 | 0.0860 | 0.294 | 0.250 | -0.045 |
| Go-ICP [14] | 150.320 | 12.261 | 2.845 | 0.112 | 0.0008 | 0.028 | 0.029 | 0.991 |
| FGR [61] | 764.671 | 27.653 | 13.794 | -3.491 | 0.0048 | 0.070 | 0.039 | 0.941 |
| PointNetLK [1] | 397.575 | 19.939 | 9.076 | -1.343 | 0.0032 | 0.0572 | 0.032 | 0.960 |
| DCP-v2 [2] | 47.378 | 6.883 | 4.534 | 0.718 | 0.0008 | 0.028 | 0.021 | 0.991 |
| PRNet (Ours) | **18.691** | **4.323** | **2.051** | **0.889** | **0.0003** | **0.017** | **0.012** | **0.995** |

Table 3: Test on unseen point clouds with Gaussian noise

Figure 1 shows the architecture of ACP. We use DGCNN with 5 dynamic *EdgeConv* layers and a Transformer to learn co-contextual representations of $\mathcal{X}$ and $\mathcal{Y}$. The number of filters in each layer of DGCNN are $(64, 64, 128, 256, 512)$. In the Transformer, only one encoder and one decoder with 4-head attention are used. The embedding dimension is 1024. We train the network for 100 epochs using Adam [63]. The initial learning rate is 0.001 and is divided by 10 at epochs 30, 60, and 80.

| Model | z/z ↑ | SO(3)/SO(3) ↑ | input size |
|---|---|---|---|
| PointNet [20] | 89.2 | 83.6 | $2048 \times 3$ |
| PointNet++ [21] | 89.3 | 85.0 | $1024 \times 3$ |
| DGCNN [22] | 92.2 | 87.2 | $1024 \times 3$ |
| VoxNet [64] | 83.0 | 73.0 | $30^3$ |
| SubVolSup [65] | 88.5 | 82.7 | $30^3$ |
| SubVolSup MO [65] | 89.5 | 85.0 | $20 \times 30^3$ |
| MVCNN 12x [66] | 89.5 | 77.6 | $12 \times 224^2$ |
| MVCNN 80x [66] | 90.2 | 86.0 | $80 \times 224^2$ |
| RotationNet 20x [67] | **92.4** | 80.0 | $20 \times 224^2$ |
| Spherical CNNs [68] | 88.9 | **86.9** | $2 \times 64^2$ |
| PRNet (Ours) | **85.2** | **80.5** | $1024 \times 3$ |

Table 4: ModelNet40: transfer learning

**Partial-to-Partial Registration on Unseen Objects.** We first evaluate on the ModelNet40 train/test split. We train on 9,843 training objects and test on 2,468 testing objects. Table 1 shows performance. Our method outperforms its counterparts in all metrics.

**Partial-to-Partial Registration on Unseen Categories.** We follow the same testing protocol as [2] to compare the generalizability of different models. ModelNet40 is split evenly by category into training and testing sets. PRNet and DCP are trained on the first 20 categories, and then all methods are tested on the held-out categories. Table 2 shows PRNet behaves more strongly than others. To further test generalizability, we train it on ShapeNetCore dataset [69] and test on ModelNet40 held-out categories. ShapeNetCore has 57,448 objects, and we do the same preprocessing as on ModelNet40. The last row in Table 2, denoted as PRNet (Ours*), surprisingly shows PRNet performs much better than when trained on ModelNet40. This supports the intuition that data-driven approaches work better with more data.

**Partial-to-Partial Registration on Unseen Objects with Gaussian Noise.** We further test robustness to noise. The same preprocessing is done as in the first experiment, except that noise independently sampled from $\mathcal{N}(0, 0.01)$ and clipped to $[-0.05, 0.05]$ is added to each point. As in Table 3, learning-based methods, including DCP and PRNet, are more robust. In particular, PRNet exhibits stronger performance and is even comparable to the noise-free version in Table 1.

## 4.2 Partial-to-Partial on Real Data

We test our model on the Stanford Bunny dataset [70]. Since the dataset only has 10 real scans, we fine tune the model used in Table 1 for 10 epochs with learning rate 0.0001. For each scan, we generate 100 training examples by randomly transforming the scan in the same way as we do in §4.1. This training procedure can be viewed as inference time fine-tuning, in contrast to optimization-based methods that perform one-time inference for each test case. Figure 2 shows the results. We further test our model on more scans from Stanford 3D Scanning Repository [71] using a similar methodology; Figure 3 shows the registration results.

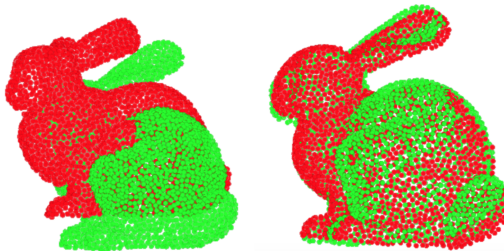

Figure 2: Left: Input partial point clouds. Right: Transformed partial point clouds.

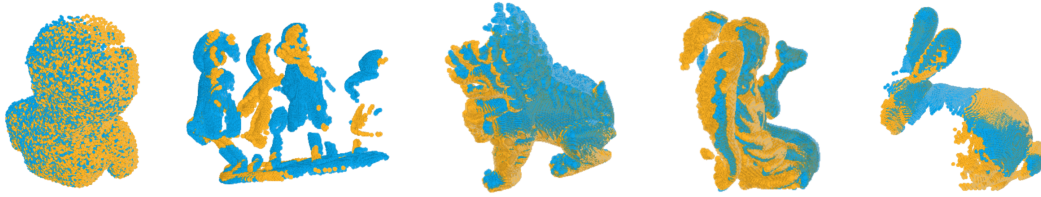

Figure 3: More examples on The Stanford 3D Scanning Repository [71].

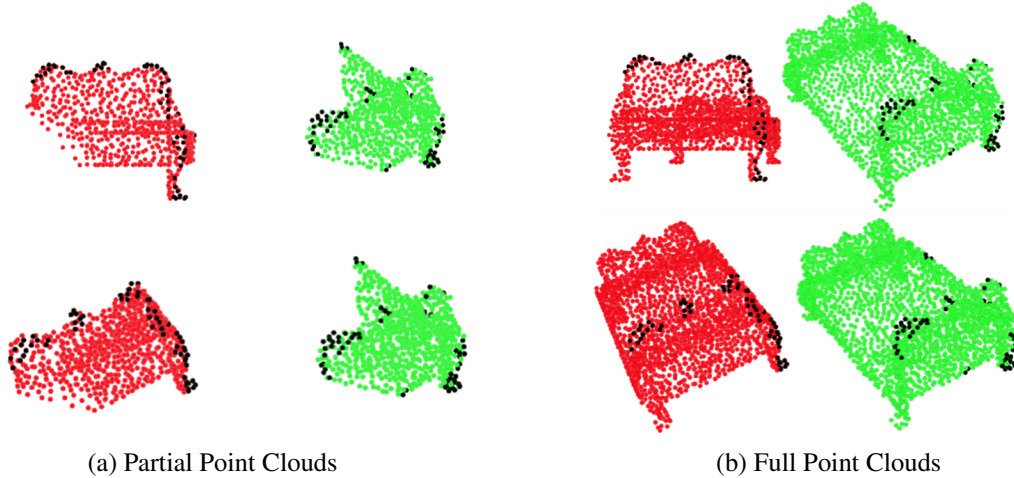

(a) Partial Point Clouds          (b) Full Point Clouds

Figure 4: Keypoint detection of a pair of beds. Point clouds in red are $\mathcal{X}$ and point clouds in green are $\mathcal{Y}$. Points in black are keypoints detected by PRNet. Point clouds in the first row are in the original pose while point clouds in the second row are transformed using the rigid transformation predicted by PRNet. (a) PRNet takes as input partial point clouds. (b) The obtained rigid transformation is applied to full point clouds for better visualization.

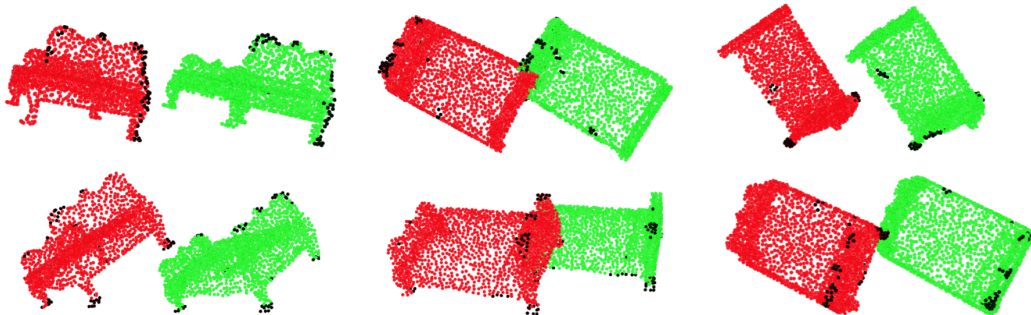

Figure 5: Keypoint detection with different partial scans. We show the same pair of $\mathcal{X}$ and $\mathcal{Y}$ with different views. The keypoints are data-dependent and consistent across different views.

## 4.3 Keypoints and Correspondences

We visualize keypoints on several objects in Figure 4 and correspondences in Figure 6. The model detects keypoints and correspondences on partially observable objects. We overlay the keypoints on top of the fully observable objects. Also, as shown in Figure 5, the keypoints are consistent across different views.

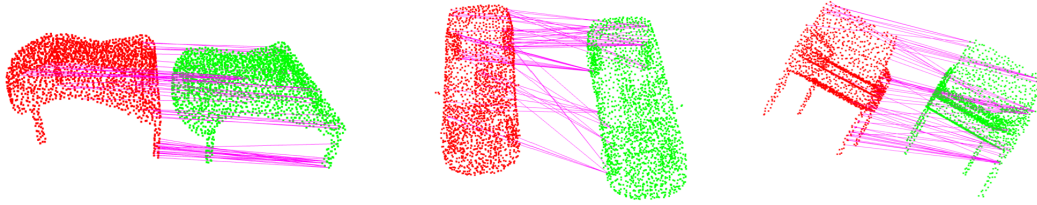

Figure 6: Correspondences for pairs of objects.

## 4.4 Transfer to Classification

Representations learned by PRNet can be transferred to object recognition. We use the model trained on ShapeNetCore in §4.1 and train a linear SVM on top of the embeddings predicted by DGCNN. In Table 4, we show classification accuracy on ModelNet40. $z/z$ means the model is trained when only azimuthal rotations are present on both during and testing. $SO(3)/SO(3)$ means the object is moved with a random motion in $SO(3)$ during both training and testing. Although other methods are supervised, ours achieves comparable performance.

## 5 Conclusion

PRNet tackles a general partial-to-partial registration problem, leveraging self-supervised learning to learn geometric priors directly from data. The success of PRNet verifies the sensibility of applying learning to partial matching as well as the specific choice of Gumbel–Softmax, which we hope can inspire additional work linking discrete optimization to deep learning. PRNet is also a reinforcement learning-like framework; this connection between registration and reinforcement learning may provide inspiration for additional interdisciplinary research related to rigid/non-rigid registration.

Our experiments suggest several avenues for future work. For example, as shown in Figure 6, the matchings computed by PRNet are not bijective, evident e.g. in the point clouds of cars and chairs. One possible extension of our work to address this issue is to use Gumbel–Sinkhorn [72] to encourage bijectivity. Improving the efficiency of PRNet when applied to real scans also will be extremely valuable. As described in §4.2, PRNet currently requires inference-time fine-tuning on real scans to learn useful data-dependent representations; this makes PRNet slow during inference. Seeking universal representations that generalize over broader sets of registration tasks will improve the speed and generalizability of learning-based registration. Another possibility for future work is to improve the scalability of PRNet to deal with large-scale real scans captured by LiDAR.

Finally, we hope to find more applications of PRNet beyond the use cases we have shown in the paper. A key direction bridging PRNet to applications will involve incorporating our method into SLAM or structure-from-motion can demonstrate its value for robotics applications and robustness to realistic species of noise. Additionally, we can test the effectiveness of PRNet for registration problems in medical imaging and/or high-energy particle physics.

## 6 Acknowledgements

The authors acknowledge the generous support of Army Research Office grant W911NF1710068, Air Force Office of Scientific Research award FA9550-19-1-031, of National Science Foundation grant IIS-1838071, from an Amazon Research Award, from the MIT-IBM Watson AI Laboratory, from the Toyota-CSAIL Joint Research Center, from a gift from Adobe Systems, and from the Skoltech-MIT Next Generation Program. Any opinions, findings, and conclusions or recommendations expressed in this material are those of the authors and do not necessarily reflect the views of these organizations. The authors also thank members of MIT Geometric Data Processing group for helpful discussion and feedback on the paper.

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
