[Supplementary Material]

| Model | MSE($\boldsymbol{R}$)↓ | RMSE($\boldsymbol{R}$)↓ | MAE($\boldsymbol{R}$)↓ | R$^2$($\boldsymbol{R}$)↑ | MSE($\boldsymbol{t}$)↓ | RMSE($\boldsymbol{t}$)↓ | MAE($\boldsymbol{t}$)↓ | R$^2$($\boldsymbol{t}$)↑ |
|---|---|---|---|---|---|---|---|---|
| Fixed $\lambda$ | 13.661 | 3.696 | 1.659 | 0.919 | 0.0003 | 0.018 | 0.012 | 0.996 |
| Annealed $\lambda$ | 12.732 | 3.568 | 1.630 | 0.924 | 0.0003 | 0.017 | 0.011 | 0.996 |
| Learned $\lambda$ | 13.506 | 3.675 | 1.675 | 0.920 | 0.0003 | 0.018 | 0.012 | 0.996 |
| Predicted $\lambda$ | **10.235** | **3.199257** | **1.454** | **0.939** | **0.0003** | **0.016** | **0.010** | **0.997** |

Table 5: Choices of $\lambda$

## Supplementary

We provide more details of PRNet in this section.

**Actor-Critic Closest Point.** A shared DGCNN [22] is to use extract embeddings for $\mathcal{X}$ and $\mathcal{Y}$ separately. The number of filters per layer are $(64, 64, 128, 256, 512)$. We use BatchNorm and LeakyReLU after each MLP in the EdgeConv layer. The local aggregation function of $k$-nn graph is max, and there is no global aggregation function used in DGCNN. $\mathcal{F}_\mathcal{X}$ and $\mathcal{F}_\mathcal{Y}$ denote the representations learned by DGCNN.

After DGCNN, $\mathcal{F}_\mathcal{X}$ and $\mathcal{F}_\mathcal{Y}$ are fed into the Transformer. The Transformer is an asymmetric function that learns co-contextual representations $\Phi_\mathcal{X}$ and $\Phi_\mathcal{Y}$. Transformer has only one encoder and one decoder. 4-head self-attention is used in encoder and decoder. LayerNorm, instead of BatchNorm, is used in the Transformer. Unlike the original implementation of Transformer, we do not use Dropout. For detailed presentation of Transformer, we refer readers to the tutorial.[1]

There are two heads on top of the representations $\Phi_\mathcal{X}$ and $\Phi_\mathcal{Y}$: a action head consisting of Gumbel-Softmax and SVD; a value head to predict a $\lambda$ for Gumbel-Softmax in the action head. The value head is parameterized by a 4-layer MLPs. The number of filters are $(128, 128, 128, 1)$. BatchNorm and ReLU are used after each linear layer in the MLPs.

**Training Protocol.** We train the model for 100 epochs. At epochs 30, 60, and 80, we divide the learning rate by 10; it is initially 0.001. Each training pair $\mathcal{X}$ and $\mathcal{Y}$ is passed through PRNet three times iteratively (the rigid alignment of $\mathcal{X}$ is updated three times). The final rigid transformation is the combination of these three local rigid transformations. $\alpha$ for cycle consistency loss and $\beta$ for feature alignment loss are both 0.1. The weight decay used is $10^{-4}$. The number of keypoints is 512 on training. For visualization purposes, however, we show 64 keypoints in Figure 4, Figure 5, and Figure 6.

Our model is trained on a Google Cloud GPU instance with 4 Tesla V100 GPUs and takes 10 hours to complete.

As for DCP-v2, we take the implementation from the authors' released code [2] and train it as they suggest.

**Choices of $\lambda$.** We compare to alternative choices of ways to determine $\lambda$: (1) fixing $\lambda$ manually; (2) annealing $\lambda$ to near 0 as the training going; (3) including $\lambda$ as a variable during training. We train the PRNet in the same way for each option, except the choice of $\lambda$ is different. Table 5 verifies our choice of strategies for computing $\lambda$.

**Keypoint detection alternatives, experiments on full point clouds, effects of discount factor, choice of $k$, robustness to data missing ratio, robustness to data noise.** To understand the effectiveness of each part, we conduct additional experiments in Table 6; to save space, we only show MAE and R$^2$. (a) First, we consider alternatives to keypoint selection: in the first alternative, the two sets of keypoints are chosen independently and randomly on the two surfaces ($\mathcal{X}$ and $\mathcal{Y}$); in the second alternative, we use *centrality* to choose keypoints, keeping the $k$ points whose average distance (in feature space) to the rest in the point cloud is minimal. Empirically, the $L^2$ norm used in our pipeline to select keypoints outperforms others. (b) Second, we compare our method to others on full point clouds. In this experiment, 768 points are sampled from each point cloud to cover the

[1] http://nlp.seas.harvard.edu/2018/04/03/attention.html
[2] https://github.com/WangYueFt/dcp

| Method | MAE($R$)↓ | R²($R$)↑ | MAE($t$)↓ | R²($t$)↑ |
|---|---|---|---|---|
| Random sampling | 1.689 | 0.927 | 0.011 | **0.997** |
| Closeness to other points | 2.109 | 0.861 | 0.013 | 0.995 |
| $L^2$ Norm | **1.454** | **0.939** | **0.010** | 0.997 |

(a) Different keypoint detection methods.

| Model | MAE($R$)↓ | R²($R$)↑ | MAE($t$)↓ | R²($t$)↑ |
|---|---|---|---|---|
| ICP | 25.165 | -5.860 | 0.250 | -0.045 |
| Go-ICP | 2.336 | 0.308 | 0.007 | 0.994 |
| FGR | 2.088 | 0.393 | **0.003** | 0.999 |
| PointNetLK | 3.478 | 0.051 | 0.005 | 0.994 |
| DCP | 2.777 | 0.887 | 0.009 | 0.998 |
| PRNet (Ours) | **0.960** | **0.979** | 0.006 | **1.000** |

(b) Experiments on full point clouds.

| Discount Factor $\lambda$ | MAE($R$)↓ | R²($R$)↑ | MAE($t$)↓ | R²($t$)↑ |
|---|---|---|---|---|
| 0.5 | 1.921 | 0.917 | 0.014 | 0.995 |
| 0.7 | 1.998 | 0.884 | 0.014 | 0.995 |
| 0.9 | **1.454** | **0.939** | **0.010** | **0.997** |
| 0.99 | 1.732 | 0.915 | 0.012 | 0.996 |

(c) Different discount factors ($\lambda$).

| Different $k$ | MAE($R$)↓ | R²($R$)↑ | MAE($t$)↓ | R²($t$)↑ |
|---|---|---|---|---|
| 16 | 27.843 | -14.176 | 0.136 | 0.326 |
| 32 | 8.293 | -1.848 | 0.048 | 0.892 |
| 64 | 3.129 | 0.563 | 0.024 | 0.979 |
| 128 | 2.007 | 0.879 | 0.016 | 0.991 |
| 256 | 1.601 | 0.932 | 0.012 | 0.996 |
| 384 | 1.508 | 0.934 | 0.011 | **0.997** |
| 512 | **1.454** | **0.939** | **0.010** | 0.997 |

(d) Different number of keypoints ($k$).

| Data Missing Ratio | MAE($R$)↓ | R²($R$)↑ | MAE($t$)↓ | R²($t$)↑ |
|---|---|---|---|---|
| 75% | 6.447 | 0.028 | 0.042 | 0.921 |
| 50% | 3.939 | 0.623 | 0.0288 | 0.969 |
| 25% | **1.454** | **0.939** | **0.010** | **0.997** |

(e) Data missing ratio.

| Data Noise | MAE($R$)↓ | R²($R$)↑ | MAE($t$)↓ | R²($t$)↑ |
|---|---|---|---|---|
| $\mathcal{N}(0, 0.01^2)$ | **2.051** | **0.889** | **0.012** | **0.995** |
| $\mathcal{N}(0, 0.1^2)$ | 5.013 | 0.617 | 0.020 | 0.991 |
| $\mathcal{N}(0, 0.5^2)$ | 21.129 | -2.830 | 0.064 | 0.917 |

(f) Data noise.

Table 6: Ablation studies.

| # points | ICP | Go-ICP | FGR | PointNetLK | DCP | PRNet |
|---|---|---|---|---|---|---|
| 512 | 0.134 | 14.763 | 0.230 | 0.049 | 0.014 | 0.042 |
| 1024 | 0.170 | 14.853 | 0.250 | 0.061 | 0.024 | 0.073 |
| 2048 | 0.242 | 14.929 | 0.248 | 0.069 | 0.058 | 0.152 |

Table 7: Inference time (in seconds).

full shape using farthest-point sampling. In the full point cloud setting, PRNet still outperforms others. (c) Third, we verify our choice of discount factor $\lambda$; small large discount factors encourage alignment within the first few passes through PRNet while large discount factors promote longer-term return. (d) Fourth, we test the choice of number of keypoints: the model achieves surprisingly good performance even with 64 keypoints, but performance drops significantly when $k < 32$. (e) Fifth, we test its robustness to missing data. The missing data ratio in original partial-to-partial experiment is 25%; we further test with 50% and 75%. This test shows that with 75% points missing, the method still achieves reasonable performance, even compared to other methods tested with only 25% points missing. (f) Finally, we test the model robustness to noise level. Noise is sampled from $\mathcal{N}(0, \sigma^2)$. The model is trained with $\sigma = 0.01$ and tested with $\sigma \in [0.01, 0.1, 0.5]$. Even with $\sigma = 0.1$, the model still performs reasonably well.

**Efficiency.** We benchmark the inference time of different methods on a desktop computer with an Intel 16-core CPU, an Nvidia GTX 1080 Ti GPU, and 128G memory. Table 7 shows learning based methods (on GPUs) are faster than non-learning based counterparts (on CPUs). PRNet is on a par with PointNetLK while being slower than DCP.

**More figures of keypoints and correspondences.** In Figure 7 and Figure 8, we show more visualizations of keypoints and correspondences for different pairs of objects.

Figure 7: Keypoint detection for different pairs of objects.

Figure 8: Correspondence prediction for different pairs of objects.