[Reviews · NeurIPS 2019]

Reviewer 1



Originality: this work tackles a traditional problem and achieves good performance improvement compared with previous states. The overall framework is quite novel. Unlike previous learning approaches which usually use a one-shot formula, the network is designed to be iterative, which is quite novel. In addition, there are also quite a few novel designs within the network. The most interesting one is the use of Gumbel-Softmax sampler within an actor-critic framework for sharp correspondence estimation. The authors have also justified the effectiveness of this through baseline comparison and ablation studies. Detecting key points for registration is also an interesting point given the main problem setup is partial scan registration. However, I feel this point is relatively weak in the whole paper with less study and justification. There's at least one work should also be mentioned and compared with, where dense correspondences between partial scans are estimated: Deep Part Induction from Articulated Object Pairs. Quality: technically, the paper is quite sound with a lot of technical details provided. However, there are still some points not clear to me. For the key point detection module, how can you guarantee the two set of top-k points are in correspondence given you are dealing with two partial scans? How does the choice of k influence the final performance? Is the key point detection module robust to the ratio of data missing? Will this l2 norm based key point detector be robust to outliers in scans? For this module design to be solid, more details should be provided. Another thing unclear is how does the discount factor influence the overall performance. Some ablation study should be provided. Clarity: the paper is clearly written and can be easily followed. Significance: point cloud registration is an important problem in computer vision and geometry processing community. The paper has done a good job in achieving state-of-the-art performance on various synthetic partial-to-partial registration tasks. The design of using Gumbel-Softmax sampler within an actor-critic framework for sharp correspondence estimation could be inspiring for other geometry processing tasks involving discrete optimization. However, all the quantitative evaluations are done with synthetic data, making the results a bit less impressive. Missing ablation studies on different design choices also weakens the overall significance a bit. Overall I like this submission but I think some points need more analysis and clarifications.

Reviewer 2



This is a very interesting paper, and its demonstration that iterative key-point methods can be learned will likely spur new research in this space. The paper provides a convincing motivation for the algorithmic approach, and strong benchmark evaluations. While the paper addresses one very specific problem in computer vision, I think the general approach to adaptive, iterative refinement of this constrained optimization problem may be of broader interest to the community. POST-REBUTTAL: Thank you for providing the ablation studies and particularly the computation speed figures. They significantly strengthen the paper.

Reviewer 3



This paper presents a novel deep network based partial-to-partial point cloud registration method named PRNet, which outperform the recently proposed DCP [2]. This paper is technically sound and the experimental results are convincing. The manuscript is well organized and well written. 3D deep learning is a hot topic in recent years, this paper goes beyond DCP and proposes a partial-to-partial registration network, which is interesting. Extensive experiments show that the proposed method achieves good performance in registration, keypoint correspondence, and transfer learning. Small issues: Can this method be used to real point cloud acquired with LIDARs mounted on mobile robots or self-driving cars? The abstract says the proposed method outperforms PointNetLK, however, the experimental results didn't include the comparison with it.

[Author Response · NeurIPS 2019]

| Method | MAE($R$)↓ | R$^2$($R$)↑ | MAE($t$)↓ | R$^2$($t$)↑ |
|---|---|---|---|---|
| Random sampling | 1.689 | 0.927 | 0.011 | **0.997** |
| Closeness to other points | 2.109 | 0.861 | 0.013 | 0.995 |
| $L^2$ Norm | **1.454** | **0.939** | **0.010** | **0.997** |

(a) Different keypoint detection methods.

| Model | MAE($R$)↓ | R$^2$($R$)↑ | MAE($t$)↓ | R$^2$($t$)↑ |
|---|---|---|---|---|
| ICP | 25.165 | -5.860 | 0.250 | -0.045 |
| Go-ICP | 2.336 | 0.308 | 0.007 | 0.994 |
| FGR | 2.088 | 0.393 | **0.003** | 0.999 |
| PointNetLK | 3.478 | 0.051 | 0.005 | 0.994 |
| DCP | 2.777 | 0.887 | 0.009 | 0.998 |
| PRNet (Ours) | **0.960** | **0.979** | 0.006 | **1.000** |

(b) Experiments on full point clouds.

| Discount Factor $\lambda$ | MAE($R$)↓ | R$^2$($R$)↑ | MAE($t$)↓ | R$^2$($t$)↑ |
|---|---|---|---|---|
| 0.5 | 1.921 | 0.917 | 0.014 | 0.995 |
| 0.7 | 1.998 | 0.884 | 0.014 | 0.995 |
| 0.9 | **1.454** | **0.939** | **0.010** | **0.997** |
| 0.99 | 1.732 | 0.915 | 0.012 | 0.996 |

(c) Different discount factors ($\lambda$).

| Different $k$ | MAE($R$)↓ | R$^2$($R$)↑ | MAE($t$)↓ | R$^2$($t$)↑ |
|---|---|---|---|---|
| 16 | 27.843 | -14.176 | 0.136 | 0.326 |
| 32 | 8.293 | -1.848 | 0.048 | 0.892 |
| 64 | 3.129 | 0.563 | 0.024 | 0.979 |
| 128 | 2.007 | 0.879 | 0.016 | 0.991 |
| 256 | 1.601 | 0.932 | 0.012 | 0.996 |
| 384 | 1.508 | 0.934 | 0.011 | **0.997** |
| 512 | **1.454** | **0.939** | **0.010** | **0.997** |

(d) Different number of keypoints ($k$).

| Data Missing Ratio | MAE($R$)↓ | R$^2$($R$)↑ | MAE($t$)↓ | R$^2$($t$)↑ |
|---|---|---|---|---|
| 75% | 6.447 | 0.028 | 0.042 | 0.921 |
| 50% | 3.939 | 0.623 | 0.0288 | 0.969 |
| 25% | **1.454** | **0.939** | **0.010** | **0.997** |

(e) Data missing ratio.

| Data Noise | MAE($R$)↓ | R$^2$($R$)↑ | MAE($t$)↓ | R$^2$($t$)↑ |
|---|---|---|---|---|
| $\mathcal{N}(0, 0.01^2)$ | **2.051** | **0.889** | **0.012** | **0.995** |
| $\mathcal{N}(0, 0.1^2)$ | 5.013 | 0.617 | 0.020 | 0.991 |
| $\mathcal{N}(0, 0.5^2)$ | 21.129 | -2.830 | 0.064 | 0.917 |

(f) Data noise.

Table 1: Ablation studies.

1 We thank reviewers for taking the time to consider our NeurIPS submission. We appreciate their feedback and will
2 revise the paper according to the comments. We also respond to some of the comments below:
3 **Keypoint detection alternatives, experiments on full point clouds, effects of discount factor, choice of $k$, robust-**
4 **ness to data missing ratio, robustness to data noise. (R1, R2)** We show results of additional experiments in Table 1;
5 to save space, we only show MAE and R$^2$. (a) First, we consider alternatives to keypoint selection: in the first
6 alternative, the two sets of keypoints are chosen independently and randomly on the two surfaces ($\mathcal{X}$ and $\mathcal{Y}$); in the
7 second alternative, we use *centrality* to choose keypoints, keeping the $k$ points whose average distance (in feature space)
8 to the rest in the point cloud is minimal. Empirically, the $L^2$ norm used in our pipeline to select keypoints outperforms
9 others. (b) Second, we compare our method to others on full point clouds. In this experiment, 768 points are sampled
10 from each point cloud to cover the full shape using farthest-point sampling. In the full point cloud setting, PRNet
11 still outperforms others. (c) Third, we verify our choice of discount factor $\lambda$; small large discount factors encourage
12 alignment within the first few passes through PRNet while large discount factors promote longer-term return. (d) Fourth,
13 we test the choice of number of keypoints: the model achieves surprisingly good performance even with 64 keypoints,
14 but performance drops significantly when $k < 32$. (e) Fifth, we test its robustness to missing data. The missing data
15 ratio in original partial-to-partial experiment is 25%; we further test with 50% and 75%. This test shows that with 75%
16 points missing, the method still achieves reasonable performance, even compared to other methods tested with only 25%
17 points missing. (f) Finally, we test the model robustness to noise level. Noise is sampled from $\mathcal{N}(0, \sigma^2)$. The model is
18 trained with $\sigma = 0.01$ and tested with $\sigma \in [0.01, 0.1, 0.5]$. Even with $\sigma = 0.1$, the model still performs reasonably well.

| | Model | MAE($R$)↓ | R$^2$($R$)↑ | MAE($t$)↓ | R$^2$($t$)↑ |
|---|---|---|---|---|---|
| Unseen point clouds | PointNetLK | 7.550 | -0.654 | 0.025 | 0.975 |
| | PRNet (Ours) | **1.454** | **0.939** | **0.010** | **0.997** |
| Unseen categories | PointNetLK | 9.655 | -2.137 | 0.033 | 0.955 |
| | PRNet (Ours) | **2.329** | **0.850** | **0.015** | **0.995** |
| With Gaussian noise | PointNetLK | 9.076 | -1.343 | 0.032 | 0.960 |
| | PRNet (Ours) | **2.051** | **0.889** | **0.012** | **0.995** |

Table 2: Comparison to PointNetLK.

| # points | ICP | Go-ICP | FGR | PointNetLK | DCP | PRNet |
|---|---|---|---|---|---|---|
| 512 | 0.134 | 14.763 | 0.230 | 0.049 | 0.014 | 0.042 |
| 1024 | 0.170 | 14.853 | 0.250 | 0.061 | 0.024 | 0.073 |
| 2048 | 0.242 | 14.929 | 0.248 | 0.069 | 0.058 | 0.152 |

Table 3: Inference time (in seconds).

**Choice of $\lambda$ in Gumbel-Softmax. (R2)** We compare to alternative choices of ways to determine $\lambda$: (1) fixing $\lambda$ manually; (2) annealing $\lambda$ to near 0 during training; and (3) including $\lambda$ as a variable during training. Table 5 in supplementary verifies our choice of computing $\lambda$. This supports the intuition that data-driven adaptive approaches usually work better.

**Alternatives to Gumbel-Softmax. (R2)** Methods to tackle non-differentiability usually fall into two categories: REINFORCE and Gumbel-Softmax. REINFORCE produces unbiased high-variance gradient estimation while Gumbel-Softmax produces biased gradients with low variance. Empirically, we

30 tried vanilla REINFORCE to estimate the gradients of the matching function; due to its instability, the training did not
31 converge. Studying unbiased low-variance gradient estimation is extremely valuable to reinforcement learning and/or
32 discrete optimization, but introducing complicated gradient estimator is beyond the scope of this paper.
33 **Comparison to PointNetLK. (R3)** Table 2 shows PRNet consistently outperforms PointNetLK in all settings. **Ef-**
34 **ficiency. (R2)** We benchmark the inference time of different methods on a desktop computer with an Intel 16-core
35 CPU, an Nvidia GTX 1080 Ti GPU, and 128G memory. Table 3 shows learning based methods (on GPUs) are faster
36 than non-learning based counterparts (on CPUs). PRNet is on a par with PointNetLK while being slower than DCP.
37 **Miscellaneous. (R1, R3)** We will add "Deep Part Induction from Articulated Object Pairs" to related works and discuss
38 about it in details. Due to time and computational resource limits, we cannot finish experiments on KITTI dataset. We
39 are actively working on extending this method to autonomous driving settings. We want to thank reviewers again for
40 providing extremely insightful and valuable feedback. We believe these comments will help to make the work stronger.

[Meta-Review · NeurIPS 2019]

This paper was well-received by the reviewers, who noted that it has a novel method with significant improvement compared to previous methods. The reviewers all requested that more analysis be performed to understand the contribution and limitations of different components of the method. It would also be helpful to show comparison to other baselines such as: Deep Part Induction from Articulated Object Pairs PointNetLK